# Current Multimodality Treatments against Brain Metastases from Renal Cell Carcinoma

**DOI:** 10.3390/cancers12102875

**Published:** 2020-10-06

**Authors:** Yoshiyuki Matsui

**Affiliations:** Department of Urology, National Cancer Center Hospital, Tokyo 104-0045, Japan; yomatsui@ncc.go.jp

**Keywords:** renal cell carcinoma, brain metastasis, targeted therapy, immune checkpoint inhibitor (ICI), whole brain radiation therapy (WBRT), stereotactic radiosurgery (SRS), prognosis, review

## Abstract

**Simple Summary:**

Brain metastasis (BM) is generally one of poor prognostic factors in patients with advanced renal cell carcinoma. However, because of longer control of extra-cranial disease by the recent introduction of molecular target therapy and immune checkpoint inhibitor, the incidence of BM has been recently increasing and to progress the treatment of BM is one of urgent medical unmet needs. Although the pivotal clinical trials usually excluded patients with BM, BM subgroup data from the prospective and retrospective series have been gradually accumulated. To select the appropriate strategy, individual patient and tumor characteristics (e.g., Karnofsky Performance Status (KPS), systemic cancer burden, the number/size/location of BM) are important information. Among the local treatments, the technology of stereotactic radiosurgery (SRT) has been especially advanced and its adaptation has been expanded. The combination of SRT with molecular target therapy and/or immune checkpoint inhibitor would be promising to further enhance the efficacy without increased toxicity.

**Abstract:**

In patients with renal cell carcinoma, brain metastasis is generally one of the poor prognostic factors. However, the recent introduction of molecular target therapy and immune checkpoint inhibitor has remarkably advanced the systemic treatment of metastatic renal cell carcinoma and prolonged the patients’ survival. The pivotal clinical trials of those agents usually excluded patients with brain metastasis. The incidence of brain metastasis has been increasing in the actual clinical setting because of longer control of extra-cranial disease. Brain metastasis subgroup data from the prospective and retrospective series have been gradually accumulated about the risk classification of brain metastasis and the efficacy and safety of those new agents for brain metastasis. While the local treatment against brain metastasis includes neurosurgery, stereotactic radiosurgery, and conventional whole brain radiation therapy, the technology of stereotactic radiosurgery has been especially advanced, and the combination with systemic therapy such as molecular target therapy and immune checkpoint inhibitor is considered promising. This review summarizes recent progression of multimodality treatment of brain metastasis of renal cell carcinoma from literature data and explores the future direction of the treatment.

## 1. Introduction

Brain metastasis (BM) is not rare among patients with advanced renal cell carcinoma (RCC); it portends a poor prognosis. Systemic therapeutic modalities for advanced RCC dramatically changed with the introduction of molecular targeted therapies and immune checkpoint inhibitors, but, as most patients with BM of RCC (RCC-BM) are excluded from important clinical trials because of poor prognosis, no validated treatment guidelines are available [1]. Localized BM treatments have also rapidly advanced with surgical and radiation technologies.

In this review, the efficacy and safety of current local, systemic, and combination treatments against RCC-BM are summarized from the latest available data, with an exploration on future directions for this unmet need.

## 2. Evidence Acquisition

### Method

A systematic literature search was carried out in the Medline, PubMed, and Web of Sciences databases from 2000 to 2020 (June), of English-language articles related to current systemic treatment of advanced RCC and its adaptation to BM and, if possible, to its efficacy and safety against RCC-BM. The treatments contain previously approved systemic therapy (cytokine therapy, molecular-targeted therapy, and immune-checkpoint inhibitor) and local therapy (surgery, whole-brain radiation therapy (WBRT), and stereotactic radiosurgery (SRS)). We also reviewed relevant clinical trials reports and reviewed reference lists from these sources for additional relevant trials.

## 3. Results

### 3.1. Trends of Systemic Therapy against Metastatic RCC (mRCC)

Most of RCC, which originates within the renal cortex, accounts for 2.4% of all malignancies diagnosed worldwide, with an estimated 403,000 new cases and 175,000 deaths globally in 2018, although there is renal medullary carcinoma, a rare entity derived from renal medulla, which is rare, aggressive, and difficult to treat and is often metastatic at the time of diagnosis [2,3,4]. The most common RCC histology is clear-cell RCC (75–85%), followed by papillary (10–15%) and chromophobe RCC (5–10%) [5]. 

Most RCC is diagnosed as organ-confined disease, which has a favorable prognosis. Unfortunately, approximately 30% of patients have metastatic disease at diagnosis or develop distant disease after treatment for local disease [6]. The most common sites of metastases include lung, bone, lymph nodes, liver, adrenal, and brain [7]. Although surgery is noncurative for mRCC, survival can vary from a few months to many years, depending on its clinical, pathologic, laboratory, and radiographic features.

The initial prognostic model of mRCC in the cytokine era, presented from the Memorial Sloan Kettering Cancer Center (MSKCC), comprises five risk factors: Performance status (PS), lactate dehydrogenase (LDH) levels, serum calcium, hemoglobin, and time from initial diagnosis to systemic treatment. Approximate median survival by the MSKCC criteria is 0 risk factors: 26 months (favorable): 1–2 risk factors: 14 months (intermediate); and three or more risk factors: 7 months (poor) [8].

Since the introduction of targeted therapies against vascular endothelial growth factor (VEGF) and mammalian target of rapamycin (mTOR) pathway in the 2000s, the International Metastatic RCC Database Consortium (IMDC) criteria, including neutrophil and platelet counts, PS, hemoglobin, calcium, and time from initial diagnosis to systemic therapy, replaced the position of the prognostic model [4]. A validation study of the IMDC system revealed median overall survival (OS) became significantly better than those seen in the cytokine era (favorable risk: 43.2 months; intermediate risk: 22.5 months; and poor risk: 7.8 months) [9,10]. Currently, VEGF-targeting agents (e.g., sunitinib, pazopanib, axitinib, tivozanib, Lenvatinib, and cabozantinib) or mTOR inhibitors (e.g., everolimus and temsirolimus) can be used to treat mRCC, either combined or as monotherapies.

More recently, the prognosis of mRCC was further improved by the introduction of immune checkpoint inhibitors (ICIs), which consist of monoclonal antibodies directed against programmed cell death 1 protein (PD-1), programmed cell death 1- ligand 1 (PD-L1), and cytotoxic T-lymphocyte antigen 4 (CTLA-4). The CheckMate-025 trial, comparing the efficacy of nivolumab (3 mg/kg) to everolimus in 821 randomized, previously treated patients, showed that median OS was longer with nivolumab (25 months) than with everolimus (19.6 months; hazard ratio [HR]: 0.73, 98.5% confidence interval [CI]: 0.57–0.93). The benefits were shown regardless of PD-L1 expression [11]. Subsequently, the CheckMate-214 trial that evaluated OS in IMDC intermediate- or poor-risk patients who were treated with ipilimumab (1 mg/kg) and nivolumab (3 mg/kg) as a combination every three weeks for four doses and then nivolumab only every two weeks, versus sunitinib (four weeks on/two weeks off schedule) showed significantly longer median OS in the ICI combination group (combination arm: Not reached; sunitinib arm: 26 months, HR: 0.63, 99.8% CI: 0.44–0.89, *p* < 0.001); the 18-month OS was also better (75% vs. 60%). Further, the nivolumab + ipilimumab arm impressively achieved Complete response (CR) in 9% of the patients compared with just 1% in the sunitinib arm, in the context of an improved Objective response rate (ORR) (nivolumab + ipilimumab: 42%, sunitinib: 27%, *p* < 0.001) [12]. Two recent randomized phase III trials showed encouraging results with a combination of ICI and tyrosine kinase inhibitor (TKI). The JAVELIN Renal 101 trial, in which 886 treatment-naïve patients with advanced clear-cell RCC (ccRCC) were randomly assigned to the combination of avelumab + axitinib versus sunitinib, showed longer median progression-free survival (PFS) (13.8 versus 8.4 months, HR: 0.69, 95% CI: 0.56–0.84) and higher ORRs (51% versus 26%). However, CR rates were similar for the two treatment arms (3% versus 2%) and the combination did not show improved OS at the data cutoff for the overall population or for any other patient subgroup (HR: 0.78, 95% CI: 0.55–1.08) [13]. The KEYNOTE-426 trial, in which 861 patients with previously untreated advanced ccRCC were randomly assigned to pembrolizumab + axitinib versus sunitinib alone, showed improved OS (18-month OS: 82% versus 72%, HR: 0.53, 95% CI: 0.38–0.74), longer median PFS (15.1 months versus 11.1 months, HR: 0.69, 95% CI: 0.57–0.84), and higher ORRs (59% versus 36%) in the combination arm [14].

As mentioned above, systemic therapy against mRCC has significantly evolved in the past two decades. Combinations of ICIs or of ICI + TKI are now standard first-line treatments as the result of several pivotal trials, although TKI monotherapy is still an important first-line therapy for those with poor PS or immunosuppression and a subsequent line for patients with refractory disease. However, patients with BM are commonly excluded from these pivotal clinical trials because of poor outcomes. We need to verify the generalizability of these standard strategies against BM.

### 3.2. Epidemiology of Brain Metastasis from RCC

Brain metastases are not a rare finding in mRCC; their five-year cumulative incidence is 7–13% [15,16,17,18,19] and has apparently risen significantly in the past two decades [20]. An estimated 2.4% of patients diagnosed with nonmetastatic RCC developed BM later; 6.5% have RCC-BM at their initial diagnoses [7,21]. Bianchi et al. found that the BM rate was significantly different according to the distribution of other noncranial metastatic sites: In case of abdominal metastases, the rate of BM was only 2%, although it reached to 16% in cases with bone, lung, and thoracic lymph nodes [7]. In the analysis of chromosomal change, the loss of chromosome 9 was reported to associate with metachronous ccRCC-BM [22].

Risk factors for RCC-BM have been widely studied. Sun et al. found that white/other race, clear-cell histology, sarcomatoid differentiation, T2-4 disease, tumor size >10 cm, and N+ disease were risk factors for BM development at RCC diagnosis [20]. Ke et al. reported that patients with tumors ≥7 cm and those younger than 70 years old had higher risk of developing BM at diagnosis [23]. Zhuang et al. suggested age 45–65 years, tumor size >10 cm, and ccRCC histology were high risk factors for BM [24]. Another report suggests that patients with pulmonary metastases and a history of tobacco abuse are more likely to harbor BM [25].

Still, current guidelines for patients with RCC do not recommend brain imaging unless BM is clinically suspected [26,27]. However, brain surveillance in the absence of symptoms, which was attributed to early detection and treatment, has been recommended in patients with advanced lung cancer and melanoma according to the guideline of the National Comprehensive Cancer Network [28]. In RCC cases, some retrospective studies have shown possible benefits from central nervous system (CNS) screening to allow identification of smaller lesions that are more amenable to treatment. Those reports suggest that those with solitary BM are less likely to develop CNS recurrence after local therapy, and selected patients with good performance status might derive prolonged survival from aggressive therapy [16,29]. 

As these advances in both local and systemic therapies provide better outcomes, patients can benefit from early detection of asymptomatic BM. Therefore, initial screening and periodic surveillance during treatment of noncranial metastasis may be worth revisiting, especially for high-risk patients with larger tumors or younger age.

### 3.3. Prognostic Factors in Patients with RCC-BM

Prognostic factors are important to determine the optimal treatment modality for RCC patients with BM. Previously, several prognostic scores for BM have been developed to determine outcome of brain surgery or radiotherapy (RT), such as the Radiation Therapy Oncology Group Recursive Partitioning Analysis (RTOG RPA), the Graded Prognostic Assessment score (GPA), the Score Index for Radiosurgery (SIR), and the Basic Score for Brain Metastases (BS-BM) [30,31,32]. However, these scores were developed and validated in studies with various cancers, but usually with few or no cases of RCC. To resolve this problem, diagnosis-specific GPA (DS-GPA) was established and RCC was selected as one of five representative cancers in this online tool, in which the score for RCC comprises Karnofsky’s PS (KPS) and the number of BM.

To improve prognostic predictions for RCC-BM further and to determine optimal treatment modalities, several recent reports have explored other prognostic factors retrospectively. Ali et al. suggested that the prognostic value of the DS-GPA scale for RCC BM was enhanced by incorporating intracranial tumor volume, in which two groups were divided by tumor volume at a 4-mL cutoff [33]. They later suggested a novel prognostic score named CERENAL, in which points were attributed according to risk factors described in RTOG RPA, BS-BM, number of brain metastases, and history of stereotactic radiosurgery [34]. Vickers et al. analyzed 106 patients with RCC-BM in the TKI era and found that KPS < 80%, diagnosis to treatment with targeted therapy <1 year, and >4 BM were associated with shorter OS since BM diagnosis [35]. Otherwise, three clinical predictors of better OS in patients with BM was reported by Bennani et al: Absence of intracranial hypertension (*p* = 0.01), deep metastases (*p* = 0.03), and systemic metastases (*p* = 0.049) [36]. Histology is still controversial. Some suggested that it is not associated with OS or local tumor control, but Takeshita et al. reported that presence of sarcomatoid components is a poor prognostic factor for patients with RCC-BM [37].

## 4. Treatment Strategy against RCC-BM

### 4.1. Local Therapy

For solitary localized and symptomatic BM with no or controlled extracranial metastasis, neurosurgery has been the gold standard treatment, especially for patients younger than 60 years, because of its rapid efficacy [38,39]. To prevent local recurrence, postsurgical radiotherapy is sometime recommended [40]. 

However, regarding WBRT and, more recently, SRS, their feasibility and safety have been also reported to improve OS, local tumor control, and clinical symptoms [41,42,43,44]. Verma et al. showed that local control of BM not only with surgery but also with SRS therapy was significantly better than no local treatment (*p* = 0.002 and *p* < 0.0001, respectively) [45]. Ippen et al. reported improved OS by combining SRS with surgery in a retrospective study of 66 patients with BM (OS after SRS only: 13.9 months, SRS with surgery: 21.9 months, and WBRT: 5.9 months) [46]. 

Especially, because of biological characteristics of RCC that is resistant to conventional fractionation RT and of WBRT-induced neurotoxicity, the use of SRS is increasing in BM management of RCC, even for multiple BM, provided that the disease burden is limited [47]. Generally, WBRT is being replaced by SRS in patients who have 1–3 BM lesions of ≤3 cm at their maximum dimension, because local control with SRS is considered to decrease as the size of the lesion increases. Commonly, lesions greater than 4 cm are considered too large for SRS. Moreover, at a threshold of 2 cm, the risks for both recurrence and radiation-induced necrosis or CNS toxicity tend to increase; both also depend on the dose, which must be decreased as volume increases [48,49,50]. However, a large study of patients with multiple BM (*n* = 1194), of whom 3% (*n* = 36) had mRCC, suggested that SRS by an experienced radiation oncologist may be safe and effective for those with up to 10 brain lesions [51].

As for dosage, SRS at a dose of 20 Gy was reported to achieve significantly better local control, in comparison with a standard dose of 16–18 Gy (local control rate at 12 months: 81% versus 50%, respectively; *p* < 0.001) [52]. From this result, high-dose, single-fraction SRS should be considered for patients who are not candidates for surgical resection, depending on the number of BM and the prognosis.

Regarding WBRT, although the indication has become limited because of its neurotoxicity, it may be still useful against multiple brain metastases (>10 metastases). Brown et al. reported the efficacy of hippocampal avoidance (HA) using intensity-modulated radiotherapy (IMRT) plus memantine, an N-methyl-D-aspartate (NMDA) receptor antagonist, to prevent cognitive dysfunction after WBRT [53]. With the use of new technology and agents, the indication of local therapy may be expanded in the future. 

### 4.2. Systemic Treatment

#### 4.2.1. Preventive Effects of Targeted Therapy on BM

Treatment with TKI agents can reduce the incidence of BM in mRCC. Massard et al. provided the first evidence from the retrospective analysis of the phase III Treatment Approaches in Renal Cancer Global Evaluation Trial (TARGET), in which those who received sorafenib had a lower incidence of BM versus placebo (12% vs. 3%) [54]. Verma et al. conducted a retrospective study to evaluate the relative impacts of sunitinib or sorafenib on BM incidence [55]; median OS periods were TKI-treated group: 25 months; no-TKI group: 12.1 months (*p* < 0.0001). Forty-four (13%) patients developed BM (no-TKI group: *n* = 29 [15.8%], TKI-treated group: *n* = 15 [9.7%]). In multivariate analysis, TKI therapy was associated with lower BM incidence (HR 0.39; 95% CI: 0.21–0.73; *p* = 0.003) and better OS (HR: 0.53; 95% CI: 0.38–0.74; *p* < 0.001). However, there was another retrospective study of 199 patients without BM, treated with or without targeted therapy (sunitinib, sorafenib, bevacizumab, temsirolimus, or everolimus) conducted by Vanhuyse et al. They reported that they could not find any significant influence of anti-angiogenic agents on the cumulative brain metastasis rate (HR: 0.67; 95% CI: 0.45–0.97; *p* = 0.18) [56]. It is still controversial, and further investigations are needed to determine the preventative effect of targeted therapy on BM.

#### 4.2.2. Efficacy and Safety of Targeted Therapy against BM

The rationale of using targeted therapy is derived from the theory that VEGF contributes to formation of cerebral lesions, although the exact mechanism of intracranial metastasis in RCC is unknown [57]. Although the main obstacle to administering drugs against BM is supposedly the blood-brain barrier (BBB), brain penetration with TKIs such as sunitinib, sorafenib, and cabozantinib has been done in a preclinical animal model [58].

The actual effect of TKI has been anecdotally reported. Lim et al. reported that sunitinib was effective in six patients with ccRCC-BM without prior local treatment, in two of whom nearly achieved CR [59]. Other studies reported the efficacy of pazopanib on BM [60,61,62]. Jacobs et al. reported a patient with BM of papillary RCC after WBRT in whom pazopanib was effective [60], and Gooch et al. reported the first case of pazopanib-responsive BM of ccRCC [57]. Another retrospective analysis of 65 patients with RCC-BM treated with targeted therapy following local treatment showed that clear cell histology (*p* < 0.0001), solitary BM (*p* = 0.004), and favorable MSKCC risk score (*p* = 0.001) are favorable prognostic factors. Neurological complications were reported in five patients in this study (hemorrhagic BM: *n* = 3, radiation-induced necrosis: *n* = 2) [63].

Two large trials are exploring the safety and efficacy of sorafenib against BM. The PREDICT (Patient characteristics in REnal cell carcinoma and Daily practICe Treatment with sorafenib) trial was a prospective, non-interventional study of open-label sorafenib to treat advanced RCC, conducted in 18 countries, to show the safety and efficacy of sorafenib in different subgroups of patients with advanced RCC. This trial included 113 patients with BM, and the median duration of sorafenib therapy was 7.0 months for those patients, which was not significantly different from the 7.3 months for the total population [64]. The ARCCS trial (The Advanced Renal Cell Carcinoma Sorafenib expanded access trial) was another open-label, nonrandomized trial that enrolled patients with advanced RCC who were not eligible for, or without access to, other sorafenib clinical trials to assess the safety and efficacy of sorafenib in the real-world setting. Of 2488 patients who helped validate its safety, 65 (2.6%) had been previously treated for BM. These patients had no CNS-related bleeding events. However, among the 47 who were evaluable for response, partial response was disappointingly reported only in two (4%), despite stable disease in 33 (70%) [65].

As for sunitinib, Gore et al. reported results from an open-label, expanded access program (EAP) for 4564 patients with mRCC from 52 countries [66]. Among 4371 patients included in the modified intention-to-treat population, 321 (7%) patients had BM at baseline. They received a median of three cycles of sunitinib (range: 1–25). Among patients with BM, 32% discontinued sunitinib for lack of efficacy and 8% because of adverse events. However, sunitinib appeared to be safe in patients with asymptomatic or previously treated BM because the incidence of toxicity was comparable to that for the overall EAP population. Of 213 evaluable patients, 26 (12%) had an objective response. Median progression-free survival (PFS) and OS were 5.6 months (95% CI: 5.2–6.1) and 9.2 months (95% CI: 7.8–10.9), respectively. One mRCC patient with BM reportedly had a complete response (CR). Their report supports the clinical activity of sunitinib in BM from RCC, but they concluded that prospective randomized trials would be required.

Based on those results, the phase II trial to prospectively evaluate intracranial response to sunitinib was started but was not completed, and finished after 16 evaluable patients, as this target objective response (OR) rate had not been reached [67]. This study explored OR in BM after two cycles of the standard regimen of sunitinib, and adopted a two-stage trial design, in which an OR rate of 35% was prospectively defined as the minimum needed to justify further investigation. Although CNS disease was stabilized in five patients, none showed OR in BM. Median time to progression was 2.3 months and OS was 6.3 months. This result led to the conclusion that sunitinib has limited efficacy and was not a good option against BM, although it is acceptably tolerable. Taken together, outcomes from first-generation TKIs were not encouraging. Therefore, no clear consensus about their use to treat BM has been reached. However, evidence about their safety would justify their use as first-line treatment in small and asymptomatic BM as part of a multidisciplinary approach and perhaps after radiotherapy in cases of stable disease.

Regarding mammallian Target Of Rapamycin (mTOR) inhibitor, the RECORD1 trial and the REACT study (RAD001 EAP) demonstrated the safety of everolimus in patients previously treated with TKIs. Those two trials included a subgroup with BM [68,69]. The safety of temsirolimus for neurologically stable RCC-BM with previous local treatment was also shown in the ARCC trial [70]. The recent METEOR trial with cabozantinib or everolimus included patients whose BM was adequately treated and had been stable for at least three months; however, data on this subgroup are not yet available [71].

#### 4.2.3. Combining TKIs with Local Therapy

Ionizing radiations increase VEGF expression and enhance expression/inhibition of other angiogenesis factors (Ang-2, Ang-1, and their receptor Tie-2) and of tumor growth factors (TGFα, Mitogen-activated Protein Kinase (MAPK)) [72,73,74,75], which might promote tumor growth and radio resistance. Logically, therefore, combining TKIs with radiation therapy (RT) to inhibit these pathways might enhance RT efficacy.

However, retrospective studies of the additional impact of TKI on local control and survival have conflicting findings [45,76,77,78,79]. Three studies retrospectively compared BM patients treated by brain RT (SRS or WBRT) with or without TKI [45,76,77]. Bates et al. suggested that concurrent TKI use was not associated with any change in OS or local control, but their analysis included only 25 patients [76]; whereas Cochran suggested that combining TKI with local therapy improved local control and extended median OS (targeted therapy: 16.6 months [*n* = 24], no targeted therapy: 7.2 months [*n* = 37]; *p* = 0.04) and lower local failure at one year (targeted therapy: 93%, no targeted therapy: 60%; *p* = 0.01) [77]. They also reported that the two groups did not significantly differ in neurological death (targeted therapy: 21.1%, no targeted therapy: 30.3%; *p* = 0.47). Verma et al. analyzed 81 patients with BM and also found that TKI was associated with a trend toward improved OS (with TKI: 6.71 months, without TKI: 4.4 months, *p* = 0.07 and significantly longer OS for patients who were TKI-naive at the time of BM development—23.6 months versus 2.08 months those who received TKI pre-BM, and 4.41 months for the never-TKI group (*p* = 0.0001) [45].

Several other retrospective studies included BM patients treated by simultaneous use of TKI with brain RT, without comparison to control groups [78,79,80,81]. Staehler et al. reported the safety and efficacy of the combination of high-dose hypofractionated RT with simultaneous targeted therapy in 51 RCC-BM cases [76]. Vickers et al. suggested that cases with KPS <80% at the start of therapy, diagnosis to treatment time <1 year, and more than four BMs were poor prognostic factors in this combination therapy from their retrospective study [35]. Because of the retrospective nature and small study population, we could not definitely conclude whether this combination improves survival. Regarding the toxicity, all the above studies showed the combination to have a good toxicity profile, which may allow continuation of this treatment when brain RT is considered. However, Kim et al. reported that the rate of radiographic radiation necrosis was significantly increased with the addition of concurrent systemic therapies, especially VEGFR-TKIs and EGFR-TKIs, to SRS and WBRT and, therefore, we must carefully follow those patients long after the combination treatments [82].

#### 4.2.4. Immune Checkpoint Inhibitors (ICI) against BM

As patients with RCC-BM have a poor prognosis, they tend to be excluded from pivotal ICI trials [12,83,84]. Additional reasons for this exclusion may depend on ICIs’ large molecular size, which limits their ability to cross the BBB; use of steroids to resolve symptomatic edema of BM, which may alter immune system activity; and the risks of metastatic pseudoprogression and hyperprogression [83,85,86].

The GETUG-AFU 26 NIVOREN phase II trial evaluated the safety and efficacy of subsequent nivolumab in patients with metastatic ccRCC after TKI failure. Eighty-five of 729 patients in this trial harbored asymptomatic BM, and detailed brain evaluation of 73 of those patients were performed within two nonrandomly assigned cohorts (the GETUG-AFU 26 NIVOREN Brain Metastases study; Table 1). Those with or without previous focal brain therapy were both included in this study; the primary endpoint was intracranial response rate in the latter cohort [87]. However, the results disappointingly showed the limited intracranial activity of nivolumab in patients with untreated BM; only four (12%) of 34 patients experienced intracranial response, and this objective response was only seen in patients with limited tumor burden (solitary tumors < 1 cm). Patients who had received prior focal therapy had significantly decreased risk of intracranial progression compared with patients with untreated BM. They concluded that CNS imaging and focal therapy should be recommended before ICI in patients with metastatic ccRCC. Several prospective trials are now ongoing to evaluate the ICI efficacy against RCC-BM, with or without local therapy (Table 2) [88].

Although the efficacy of single ICI treatment may be limited in BM, antitumor activity from combination ICIs has been observed in patients with BM from melanoma treated with nivolumab (1 mg/kg) + ipilimumab (3 mg/kg) and in patients with non-small cell lung cancer treated with nivolumab (240 mg) + ipilimumab (1 mg/kg). Unfortunately, CheckMate 214, which evaluated the combination of nivolumab and ipilimumab as first-line therapy against mRCC, excluded those with BM [12], but the ongoing CheckMate 920 phase 3b/4 clinical trial is including those with brain metastasis, non-ccRCC, and poor KPS. The primary endpoint of this trial is to evaluate the safety of nivolumab + ipilimumab combination treatment by the incidence of high-grade, immune-mediated adverse events (IMAEs) in such high unmet medical needs (Table 1), but the efficacy was also examined by PFS and ORR as key secondary endpoints [88]. Overall, 28 RCC-BM patients (KPS ≥ 70%) were enrolled in this study, and six patients (21.4%) reportedly had grade 3-4 IMAEs within 100 days of their last doses during a minimum follow-up of 6.47 months. As for the efficacy, ORR was 28.6% (95% CI: 13.2–48.7) and median PFS was 9.0 months (95% CI: 2.9—not estimable [NE]), indicating that the ICI combination treatment may also be promising in RCC patients with BM, with an acceptable safety profile.

Another first-line combination option is ICI+TKI, but the pivotal trials, such as KEYNOTE-426 and JAVELIN Renal 101, also excluded those with active CNS metastases [13,14]. However, 46 patients enrolled in JAVELIN Renal 101 trial had BM and had completed their treatments and recovered from the acute effects of radiation therapy or surgery prior to randomization (23 in each arm). Jonasch et al. performed subgroup analysis comparing those patients with those without BM, reporting that patients assigned to avelumab + axitinib had a PFS of 4.9 months (95% CI: 1.6–5.7) vs. 2.8 months (95% CI: 2.3–5.6) for patients assigned to sunitinib (control; HR: 0.90; 95% CI: 0.43–1.88) (Table 1). The authors concluded that the observed PFS among patients with BM at enrollment was similar between the two arms, with a hazard ratio and medium PFS that numerically favored the avelumab arm. However, the prognosis of those patients is still poor and more effective treatments are needed [89].

### 4.3. Combining ICI with Radiation Therapy

Recent preclinical studies have suggested a synergy between radiation and immunotherapies [90,91,92,93], especially for concurrent ICI with RT [90], possibly because RT evokes immunological changes in both the tumor and its microenvironment (by promoting effector immune cell recruitment), and might induce systemic responses by promoting antitumor immunity (the “abscopal effect”), via several mechanisms, such as enhanced tumor antigen release, exposure of novel tumor antigens, increased immunogenic cell death, and increased pro-inflammatory cytokines that activate T cells [94]. Through its action, particularly in the tumor microenvironment, RT might facilitate immunotherapies such as ICI. 

Most available clinical studies to evaluate the efficiency of RT and ICI combinations are retrospective [95,96,97,98], and their cohorts were mostly patients with melanoma BM treated with ipilimumab and RT (mainly SRS). These studies suggested longer OS with the combined treatment, without increased toxicity.

Several studies have analyzed the optimal timing of the combination. A retrospective study of 75 melanoma patients with BM by Qian et al. suggested that administering ICI within four weeks of SRS improved the local response compared with initiating treatment more than four weeks later [99]. Furthermore, anti-PD-1 therapy also resulted in greater local response than anti-CTLA-4 after SRS. They concluded that immunotherapy can have a synergistic effect with radiosurgery in BM treatment, even in those not known to have PD-L1 expression, and that early local response is greater and more rapid with concurrent immunotherapy and SRS. Chen also retrospectively investigated the optimal timing of ICI and RT, with 260 patients, of whom 33 had RCC [100]. They reported that concurrent ICI (within two weeks) was not associated with increased rates of immune-related adverse events or acute neurologic toxicity and predicted a decreased likelihood of developing ≥3 new BMs after SRS. SRS with concurrent ICI was associated with improved OS compared with SRS alone (*p* = 0.002; HR: 2.69) and non-concurrent SRS and ICI (*p* = 0.006; HR: 2.40) in multivariate analysis. The improvement in OS from concurrent SRS and ICI was significant compared with patients treated with SRS before ICI (*p* = 0.002; HR: 3.82) or after ICI (*p* = 0.021; HR: 2.64). However, regarding delayed radiation neurotoxicity, intratumor hemorrhage, and edema, Martin et al. found an association between immunotherapy and symptomatic radiation necrosis in patients who underwent SRS for BM from melanoma, non-small cell lung cancer, or RCC [101]. This association was especially strong in patients with melanoma rather than RCC (HR: 4.02; 95% CI: 1.17–13.82; *p* = 0.03), but only five patients with RCC in this study were by this combination. Completion of further prospective trials with longer follow-up is awaited to validate the efficacy and safety of combined ICI and SRS for RCC-BM. 

## 5. Conclusions

The prognosis for RCC patients with BM is generally poor, but aggressive multimodality treatment can give some patients prolonged survival. To select the appropriate approach, individual patient and tumor characteristics (e.g., KPS status, systemic cancer burden, the number/size/location of BM) would be important information. Most recent studies apparently support TKIs or ICI combined with SRS against BM, without altering the clinical safety profiles. However, further prospective investigations with longer follow-up are needed to confirm the retrospective data that indicates the efficacy and safety of aggressive combination treatments. Moreover, although most trials have been targeting the patients with ccRCC, the treatment against brain metastases from non-ccRCC is also one of unmet needs, which should be resolved in the future.

## Figures and Tables

**Table 1 cancers-12-02875-t001:** The comparison of the results of clinical trials of ICI against RCC between whole group and brain metastases subgroup.

≥Second Line Nivolumab	Phase	*n*	Endpoint	Results		Remarks
GETUG-AFU 26 NIVOREN	2	729	median PFS	3.2 months (95% CI 2.9–4.6)		
			1y OS	69% (95% CI 66–73)		
			ORR	21%		
GETUG-AFU 26 NIVOREN Brain Metastases study		85		prior focal therapy		
				+	‒	
			ORR	NA	12%	
			median PFS	4.8 mo	2.7 mo	
			1y OS	59%	67%	
nivolumab + ipilimumab vs. sunitinib CheckMate 214	3	1096		NIVO + IPI	sunitinib	
			18 mo OS	75%	60%	
			30 mo PFS	28%	12%	
			median OS	not reached	37.9 mo	HR, 0.71; 95% CI, 0.59–0.86; *p* = 0.0003
			ORR	41%	34%	*p* = 0.015
CheckMate 920 brain metastases cohort	3b/4	28	Median OS	not reached (95% CI 13.1–NE)		
			ORR	28.6% (95% CI 13.2–48.7)		
			immune-mediated adverse events (IMAEs)	6 (21.4%) (G3-4: 1 pt)		
avelumab + axitinib vs. sunitinib JAVELIN Renal 101	3	886		AVE + AXI	sunitinib	
				442	444	
			Median PFS	13.8 mo	7.2 mo	HR: 0.61; 95% CI: 0.47, 0.79; *p* < 0.001
			ORR	51.4 (46.6–56.1)	25.7 (21.7–30.0)	
			BM develop	8	10	
JAVELIN Renal 101 subgroup with Brain metstases	-	46		23	23	
			median PFS	4.9 mo	2.8 mo	HR: 0.90; 95% CI: 0.43, 1.88

ICI = immune checkpoint inhibitor, RCC = renal cell carcinoma, ORR = objective response rate, OS = overall survival, PFS = progression-free survival, BM = brain metastasis, NA = not assessed.

**Table 2 cancers-12-02875-t002:** Summary of the ongoing trials of TKI or ICI in brain metastases of RCC.

Drugs	Trial	Phase	Systemic Treatment	Local Therapy	Population	Estimated Enrollment	Primary Endpoint	Status
TKI	NCT00981890	1	sunitinib	SRS	brain metastasis	22	safety and maximum tolerated dose of sunitinib	Active, not recruiting
	NCT02019576	2	sunitinib	SRS	ccRCC	68	local control at 1 yr of metastases treated with SRS	Active, not recruiting
ICI	NCT02886585	2	pembrolizumab	(SRS)	solid tumor	102	ORR, OS, Extracranial ORR	Recruiting
	NCT02978404	2	nivolumab	SRS	NSCLC, RCC	60	Intracranial PFS	Active, not recruiting
	NCT02982954	4	nivolumab/ipilimumab	-	aRCC	200	incidence of IMAEs	Active, not recruiting
	NCT02669914	2	durvalumab	WBRT/SRS	epithelial-derived tumor	136	ORR of Intracranial Disease	Terminated (Low accrual)

TKI = tyrosin kinase inhibitor, ICI = immune checkpoint inhibitor, SRS = stereotactic radiosurgery, WBRT = whole brain radiotherapy, RCC = renal cell carcinoma, NSCLC = non-small cell lung cancer, ccRCC = clear-cell RCC, aRCC = advanced RCC, ORR = objective response rate, OS = overall survival, PFS = progression-free survival, IMAE = immune-mediated adverse event.

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
