# Peer review of "Current Multimodality Treatments against Brain Metastases from Renal Cell Carcinoma"

_cancers, 2020, doi:10.3390/cancers12102875_

Round 1

Reviewer 1 Report

The review paper by Dr. Matsui deals with the current treatment against brain metastases from renal cell carcinoma. I think, most of the hot topics were comprehensively and well covered, but I recommend the author to address the following issues listed below.

  1. RCC does not originate only within the renal cortex, but also medulla. Renal medullary carcinoma is derived from the renal medulla and is a rare, aggressive, difficult to treat, and is often metastatic at the time of diagnosis. Please re-consider the general information written in line 57.
  2. The authors wrote about ccRCC only in most of the issues. It will be more useful if authors can discuss the current treatment against brain metastases from non-clear cell RCC for several histological subtypes.

Reviewer 2 Report

This is a well-written and important review article.

I have only one relevant comment:

In chapter 4.2.3. (Combining TKIs with local therapy), some information regarding the toxicity of combinations of TKIs and radiosurgery/FSRT should be added, either using the studies cited in this chapter or interpolating data from studies reporting on brain mets from another less-radiosensitive tumor, namely melanoma (several studies and case reports available in the literature). It should become clear that the combination of TKIs and SRS bears a significant risk of necrosis of healthy brain tissue.       

Reviewer 3 Report

This manuscript provides a good summary of the current evidence in management of brain metastases from renal cell carcinoma. The article covers the most recent data in terms of systemic treatment options for these patient.

Major suggestion:

Local Therapy: I recommend adding that there is still a role for WBRT in patients with multiple (>10 brain metasetases). Also the readers might be interested in the recent study by Brown et al. on hippocampal avoidance during WBRT plus memantine. (PMID: 32058845)

Minor suggestion:

Line 131-132: Please add citation where evidence has showed improvement in OS with brain surveillance in lung and breast cancer.
